# Hyperactive Natural Killer cells in *Rag2* knockout mice inhibit the development of acute myeloid leukemia

Emi Sugimoto[1,9], Jingmei Li[2,9], Yasutaka Hayashi[1], Kohei Iida[2], Shuhei Asada [1], Tsuyoshi Fukushima[1], Moe Tamura[2], Shiori Shikata[2], Wenyu Zhang[2], Keita Yamamoto[2], Kimihito Cojin Kawabata[1], Tatsuya Kawase[3], Takeshi Saito[4], Taku Yoshida[3], Satoshi Yamazaki [5], Yuta Kaito[6], Yoichi Imai[7], Tamami Denda[8], Yasunori Ota[8], Tomofusa Fukuyama [1], Yosuke Tanaka [1], Yutaka Enomoto[1], Toshio Kitamura [1✉] & Susumu Goyama [2✉]

Immunotherapy has attracted considerable attention as a therapeutic strategy for cancers including acute myeloid leukemia (AML). In this study, we found that the development of several aggressive subtypes of AML is slower in $Rag2^{-/-}$ mice despite the lack of B and T lymphocytes, even compared to the immunologically normal C57BL/6 mice. Furthermore, an orally active p53-activating drug shows stronger antileukemia effect on AML in $Rag2^{-/-}$ mice than C57BL/6 mice. Intriguingly, Natural Killer (NK) cells in $Rag2^{-/-}$ mice are increased in number, highly express activation markers, and show increased cytotoxicity to leukemia cells in a coculture assay. B2m depletion that triggers missing-self recognition of NK cells impairs the growth of AML cells in vivo. In contrast, NK cell depletion accelerates AML progression in $Rag2^{-/-}$ mice. Interestingly, immunogenicity of AML keeps changing during tumor evolution, showing a trend that the aggressive AMLs generate through serial transplantations are susceptible to NK cell-mediated tumor suppression in $Rag2^{-/-}$ mice. Thus, we show the critical role of NK cells in suppressing the development of certain subtypes of AML using $Rag2^{-/-}$ mice, which lack functional lymphocytes but have hyperactive NK cells.

[1] Division of Cellular Therapy, Institute of Medical Science, The University of Tokyo, Tokyo, Japan. [2] Division of Molecular Oncology, Department of Computational Biology and Medical Sciences, Graduate School of Frontier Sciences, The University of Tokyo, Tokyo, Japan. [3] Drug Discovery Research, Astellas Pharma, Ibaraki, Japan. [4] Clinical Pharmacology Exploratory Development, Astellas Pharma, Westborough, MA, USA. [5] Laboratory of Stem Cell Therapy, Faculty of Medicine, University of Tsukuba, Ibaraki, Japan. [6] Department of Hematology/Oncology, IMSUT Hospital, The University of Tokyo, Tokyo, Japan. [7] Department of Hematology and Oncology, Dokkyo Medical University, Tochigi, Japan. [8] Department of Pathology, The Institute of Medical Science Research Hospital, The University of Tokyo, Tokyo, Japan. [9] These authors contributed equally: Emi Sugimoto, Jingmei Li. ✉email: kitamura@ims.u-tokyo.ac.jp; goyama@edu.k.u-tokyo.ac.jp

Acute myeloid leukemia (AML) is an aggressive form of hematological malignancy with uncontrolled over-production of myeloid cells[1]. Hematopoietic stem cell transplantation (HSCT) is currently the only curative therapy for AML owing to the graft versus leukemia (GVL) effect, which is mainly mediated by alloreactive donor T cells[2,3]. T cells represent a major component of the adaptive immune system. Cytotoxic CD8[+] T cells recognize tumor cells through major histocompatibility complex class I (MHC-I) with peptides from aberrant cancer specific proteins. Recently, innovative T-cell-based cancer immunotherapies, such as chimeric antigen receptor (CAR)-T cells, bispecific T-cell engagers (BiTEs) and checkpoint inhibitors have been developed[4,5]. Although these therapies that harness anticancer-cell activity have shown stellar success in the treatment of solid and B-cell tumors, few have reached clinical use yet for AML.

In addition to the adaptive immune system, increasing evidence supports a role for innate immune effector cells, such as Natural killer (NK) cells, in tumor surveillance[6]. NK cells are innate lymphoid cells that control viruses and tumors through cytotoxicity and cytokine production. The killer-cell immunoglobulin-like receptors on NK cells interact with MHC-I on target cells, which inhibits NK cell function. Therefore, NK cells preferentially kill cells lacking self-MHC-I expression. Given that MHC-I downregulation is an important mechanism for evading T-cell immune surveillance, NK cells can complement T-cell immunity to eradicate T-cell-resistant tumors with low MHC-I expression. NK cell activity is also regulated by multiple inhibitory (e.g., CD94/NKG2A) and activating receptors (e.g., CD94/NKG2D)[6]. Interestingly, a recent study showed that AML cells that do not express NKG2D ligands have the stemness characteristics[7], indicating that leukemia stem cells (LSCs) in AML can escape the NK cell immune recognition. The anti-tumor activity of NK cells was first demonstrated by Ruggeri et al. who observed that AML patients whose donor-derived NK cells had alloreactivity in GVL direction showed a significantly superior survival[8]. Since then, NK cell-based immunotherapy, such as adoptive NK cell transfer and CAR-NK cells, have attracted considerable attention as promising options to treat AML[9].

To understand the role of functional immune system in AML progression, we have used several mouse myeloid tumor models driven by MLL-AF9, RUNX1-ETO9a, ASXL1/RUNX1 mutations and ASXL1/SETBP1 mutations. MLL-AF9 is a fusion gene resulting from the chromosomal translocation t(9;11)(p22;q23) and has the potent leukemogenic potential to transform hematopoietic progenitor cells into AML cells[10,11]. RUNX1-ETO is one of the most frequent cytogenetic abnormalities in AML, resulting from the chromosomal translocation t(8;21)(q22;q22)[12]. RUNX1-ETO9a, a splice variant of RUNX1-ETO with stronger leukemogenic potential, can transform mouse bone marrow cells into AML cells[13]. ASXL1 is an epigenetic regulator and the mutations in *ASXL1* gene often coexist with *RUNX1* and *SETBP1* mutations in myelodysplastic syndromes (MDS) and AML[14]. We transduced these oncogenes into mouse bone marrow progenitor cells, transplanted them into recipient mice, and produced AML or MDS/AML in mice. The oncogene-driven mouse AML and MDS/AML cells were then enriched for leukemia stem cell activity by serial transplantations. These engineered myeloid tumors can engraft and initiate leukemia even in non-irradiated recipient mice with functional immune system, which allows us to assess the role of tumor immunity in myeloid leukemogenesis.

Using the MLL-AF9-driven AML model, we previously showed that checkpoint inhibitors augmented the therapeutic effects of the p53-activating drug on AML[15]. However, the relative importance of T/B lymphocytes and NK cells during AML progression remained unclear. To address this question, we compared the latencies of MLL-AF9-induced AML in immunologically normal wild-type (WT) C57BL/6 mice, *Rag2*[−/−] mice lacking mature T and B cells, and NSG mice lacking T, B and NK cells[16]. This experiment led to the intriguing discovery that the development of AML was inhibited in *Rag2*[−/−] mice that lack T/B lymphocytes but have hyperactive NK cells.

## Results

**Unexpected delay of AML development in *Rag2*[−/−] mice.** We first established a mouse model for MLL-AF9 leukemia. We transduced MLL-AF9 (coexpress GFP) into mouse bone marrow progenitor cells, and transplanted the cells into sublethally irradiated recipient mice. All recipient mice developed AML around 2 months after transplantation. We collected GFP[+] leukemia cells from spleens of the moribund mice, and enriched for leukemia stem cell activity by serial transplantation through secondary, tertiary, and quaternary recipient mice after sublethal irradiation. The MLL-AF9-expressing AML cells were then maintained in non-irradiated recipient mice through serial (more than 10 times) transplantations. These aggressive MLL-AF9 cells were then transplanted into WT, *Rag2*[−/−] or NSG mice to determine the role of specific immune cells in the development of AML (Fig. 1a). Consistent with our previous results[15], we observed earlier onset of AML in the immunodeficient NSG mice. Surprisingly, AML progression was significantly slower in *Rag2*[−/−] mice even compared to WT mice with normal immune function (Fig. 1b). Engraftment of AML cells in peripheral blood was profoundly reduced in *Rag2*[−/−] mice (Fig. 1c), and the splenomegaly induced by the infiltrated MLL-AF9 cells was not evident in *Rag2*[−/−] mice (Fig. 1d). Thus, the development of AML was slowed in *Rag2*[−/−] mice despite the lack of mature T and B lymphocytes. We then assessed the effect of the p53-MDM2 interaction inhibitor, DS-5272[17], on survival time in WT and *Rag2*[−/−] mice transplanted with MLL-AF9 cells. Strikingly, DS-5272 treatment showed drastic antileukemia effect in *Rag2*[−/−] mice and cured most of leukemia-bearing mice, which was rarely achieved in WT mice (Fig. 1e). Thus, *Rag2*[−/−] mice have the unexpected tumor-suppressive environment for MLL-AF9-induced AML. Given that several studies have shown the antitumor effect of NK cells against MLL-fusion leukemia[18,19], these results also suggest the involvement of NK cells, which are retained in *Rag2*[−/−] mice, in suppressing the development of AML.

**MHC-I depletion suppresses MLL-AF9-driven leukemogenesis in WT and *Rag2*[−/−] mice.** Next, we assessed expression profiles of MLL-AF9 cells collected from moribund WT mice, *Rag2*[−/−] mice and NSG mice. Interestingly, AML cells from *Rag2*[−/−] mice showed distinct gene expression profile compared to those from other mice, including upregulation of genes related to innate immune and inflammatory responses (Fig. 2a–c, Supplemental Fig. 1, Supplementary Data 1). In accordance with the enhanced inflammation, surface expression of MHC-I (H2-Kb) was upregulated in AML cells from *Rag2*[−/−] mice (Fig. 2d). Other ligands for NK cell receptors, Qa-1b, Rae1, Mult1, were not strongly expressed in MLL-AF9 cells (Fig. 2e). These data indicate that MHC-I upregulation driven by the increased inflammatory signal may provide survival benefit to AML cells specifically in *Rag2*[−/−] mice. To test this possibility, we assessed the role of B2m, a component of MHC-I, in MLL-AF9-induced leukemogenesis. We transduced Cas9 together with a tRFP657-coexpressing non-targeting (NT)-single guide RNA (sgRNA) or two independent sgRNAs targeting *B2m* into MLL-AF9 cells. These cells were cultured in vitro or directly transplanted into WT, *Rag2*[−/−] and NSG mice (Fig. 3a). Decrease of H2-Kb expression in the B2m-targeting sgRNA-transduced cells was confirmed by FACS

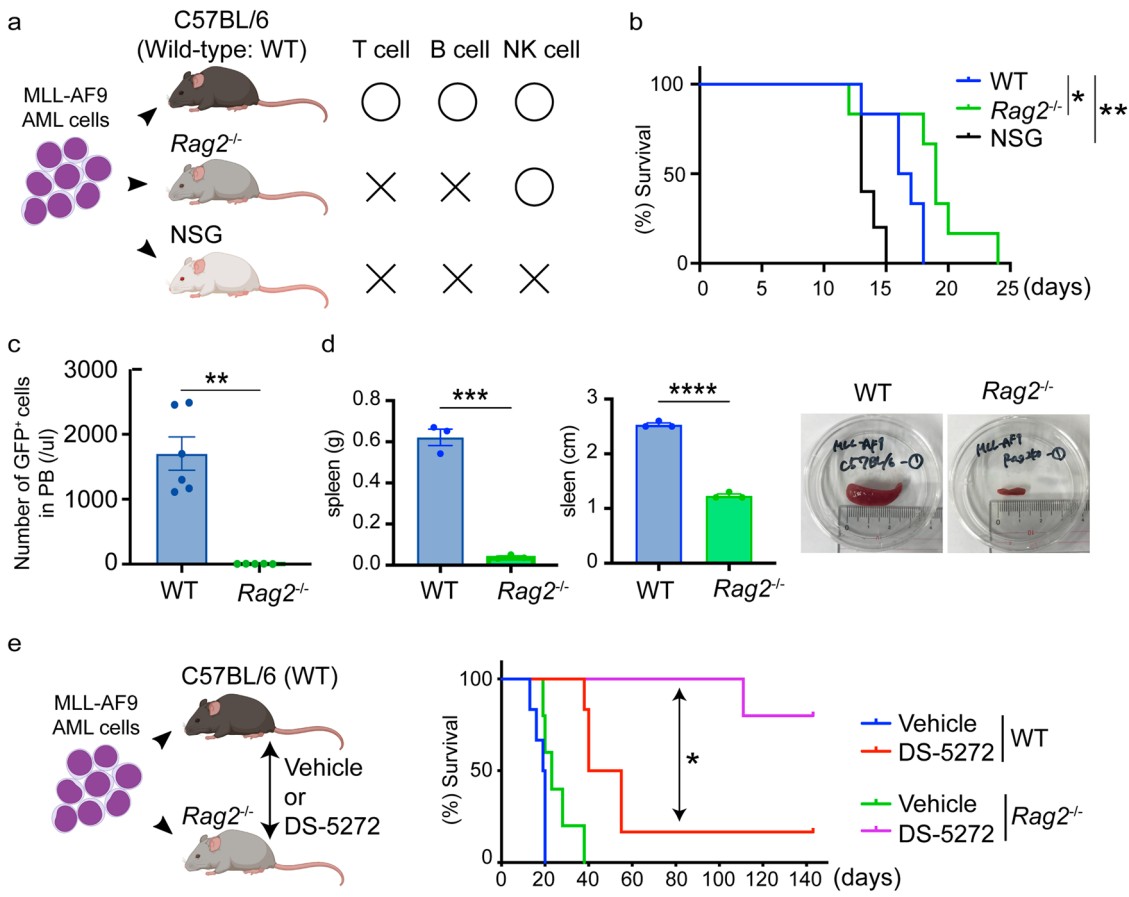

**Fig. 1 Delayed development of MLL-AF9-induced AML in $Rag2^{-/-}$ mice. a** Experimental scheme used in **b–d**. Mouse bone marrow cells were transformed by MLL-AF9 and enriched for leukemia stem cell activity by serial transplantations (more than 10 times). The MLL-AF9 cells with strong leukemogenicity were then transplanted into wild-type (WT) C57BL/6 mice, $Rag2^{-/-}$ mice, or NSG mice. Figures were created with BioRender (https://app.biorender.com/). **b** Kaplan–Meier survival curves of female WT, $Rag2^{-/-}$ or NSG mice transplanted with MLL-AF9 cells are shown. (WT: $n = 6$, $Rag2^{-/-}$: $n = 6$, NSG: $n = 5$). *$P < 0.05$, **$P < 0.01$; log-rank test. **c** Number of GFP+ MLL-AF9 cells in peripheral blood of WT and $Rag2^{-/-}$ mice. PB was collected 11 days after transplantation. Data are shown as means ± s.e.m. [$n = 6$ (male: $n = 3$, female: $n = 3$) for each group]. **$P < 0.01$ ; two-tailed Mann–Whitney test. **d** Weight (left) and size (middle) of spleens collected from WT or $Rag2^{-/-}$ mice 11 days after transplantation. Representative pictures of the spleen were also shown (right). Data are shown as means ± s.e.m. ($n = 3$ for each group, male). ***$P < 0.001$, ****$P < 0.0001$; two-tailed Student's $t$ test **e** Kaplan–Meier survival curves of MLL-AF9-bearing WT or $Rag2^{-/-}$ mice treated with vehicle control or DS-5272 (100 mg/kg, three times weekly for 2 weeks, starting 3 days after transplantation). [WT/Vehicle: $n = 6$ (male/female $= 3/3$), WT/DS-5272: $n = 6$ (m/f $= 3/3$), $Rag2^{-/-}$/Vehicle: $n = 5$ (m/f $= 2/3$), $Rag2^{-/-}$/DS-5272: $n = 5$ (m/f $= 3/2$)]. *$P < 0.05$; log-rank test. Figures (left) were created with BioRender (https://app.biorender.com/).

(Fig. 3b). Although B2m depletion did not change the growth of MLL-AF9 cells in vitro and in NSG mice, it inhibited AML progression in WT and $Rag2^{-/-}$ mice, as evidenced by the decrease of tRFP657+ cells (B2m-sgRNA-transduced cells) (Fig. 3c, d). We also found that most tRFP657+ cells expressed H2-Kb in WT and $Rag2^{-/-}$ mice, while those in NSG mice lost H2-Kb expression (Fig. 3e). The data indicate that MLL-AF9 cells cannot grow well in the absence of MHC-I expression in WT and $Rag2^{-/-}$ mice, and it is likely that a small fraction of MLL-AF9 cells that escaped sgRNA-mediated B2m excision expand and predominate over B2m-deficient leukemia cells. Given that loss of MHC-I is known to enhance the NK cell-mediated cytotoxicity, these results suggest the important role of NK cells to inhibit the development of MLL-AF9-induced AML. Consistent with these findings, a co-culture assay with NK cells and MLL-AF9 cells revealed that B2m depletion enhanced NK cell-mediated anti-leukemic effect against MLL-AF9 cells (Fig. 3f).

**Hyperactive NK cells inhibit AML progression in $Rag2^{-/-}$ mice.** Based on the above results, we hypothesized that NK cell-driven tumor immunity in $Rag2^{-/-}$ mice has a stronger anti-tumor effect

than that in WT mice. Indeed, both the number and frequency of NK cells were increased in $Rag2^{-/-}$ mice compared to WT mice in peripheral blood, spleen and bone marrow (Fig. 4a, Supplemental Fig. 2a). In addition, NK cells in $Rag2^{-/-}$ mice expressed higher levels of activation markers[20], CD107a, CD69 and Sca-1, than those in WT mice (Fig. 4b, Supplemental Fig. 2b). To assess the cytotoxic activity of NK cells, we isolated NK cells from WT and $Rag2^{-/-}$ mice and co-cultured them with a mouse lymphoma cell line YAC-1. The co-culture assay revealed that $Rag2^{-/-}$ NK cells expressed higher level of CD107a in the presence of YAC-1 cells and induced apoptosis of YAC-1 cells more efficiently than WT NK cells (Fig. 4c, d). Furthermore, NK cells derived form $Rag2^{-/-}$ mice produced more IFNγ upon stimulation with PMA and ionomycin than those from WT mice (Fig. 4e). Histological analyses of bone marrows and spleens also revealed the increase of NK cells in $Rag2^{-/-}$ mice transplanted with MLL-AF9 cells (Fig. 4f, Supplemental Fig. 2c). Thus, all these results indicate the increased activity of NK cells in $Rag2^{-/-}$ mice. To determine the role of NK cells in $Rag2^{-/-}$ mice during AML progression, we then depleted NK cells with the anti-NK1.1 antibody from WT and $Rag2^{-/-}$ mice transplanted with MLL-AF9 cells (Fig. 4g). The survival

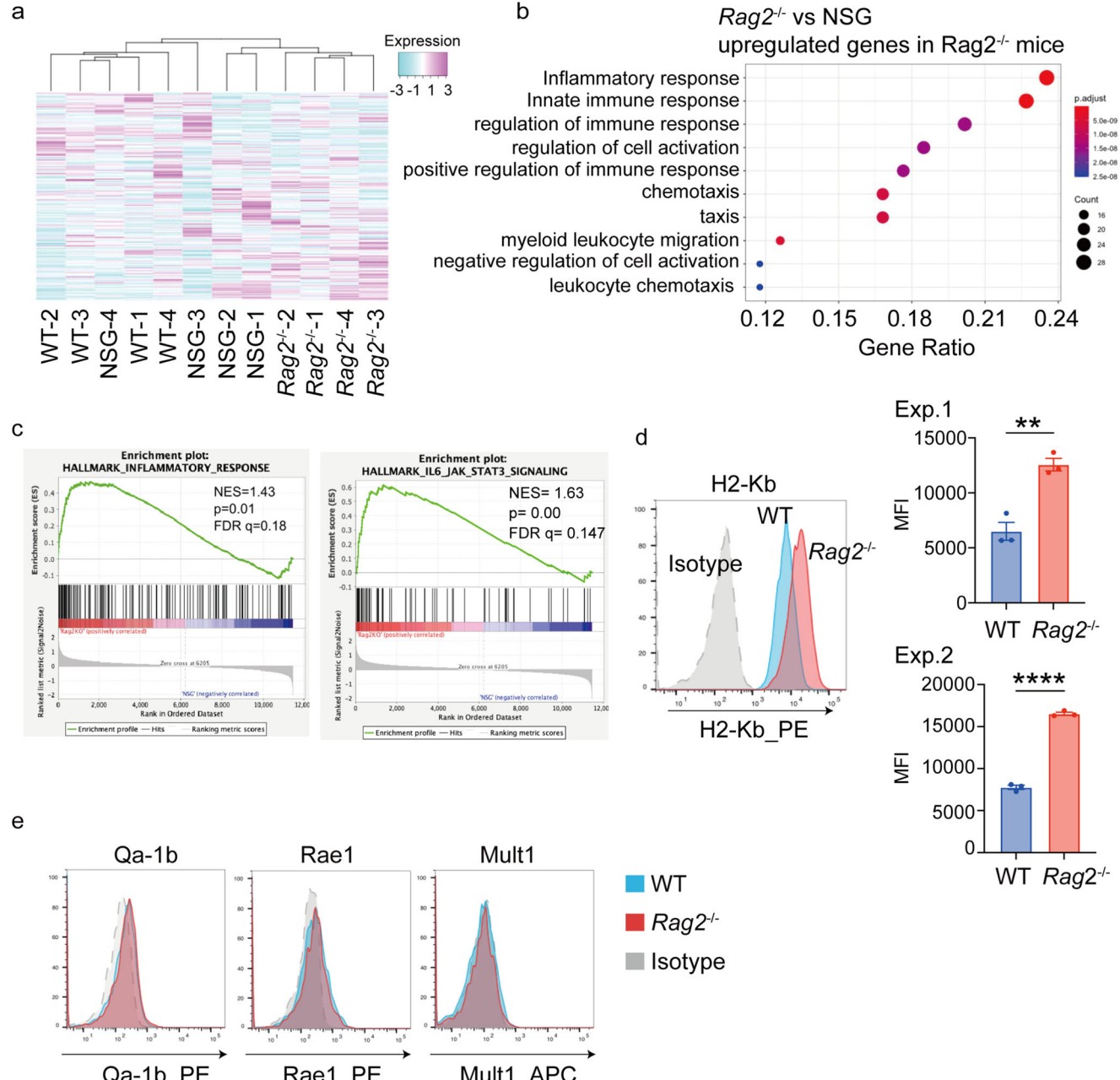

**Fig. 2 Increased expression of MHC class I in AML cells in $Rag2^{-/-}$ mice. a** Hierarchical clustering of the RNA-Seq data of MLL-AF9 cells collected from female WT, $Rag2^{-/-}$ or NSG mice, showing distinct expression pattern of $Rag2^{-/-}$ mice-derived MLL-AF9 cells ($n = 4$ for each group). **b** GO analysis of upregulated genes in MLL-AF9 AML cells collected from $Rag2^{-/-}$ mice compared with those from NSG mice. Inflammation- and immune-related genes were upregulated in MLL-AF9 cells collected from $Rag2^{-/-}$ mice. See also Supplemental Fig. 1. **c** GSEA of upregulated genes in MLL-AF9 AML cells collected from $Rag2^{-/-}$ mice compared with those from NSG mice, showing the enrichment of genes related to the inflammatory response and IL-6-JAK-STAT3 pathway. **d** Expression of MHC class I (H2-Kb) on MLL-AF9 cells collected from male WT or $Rag2^{-/-}$ mice. Representative histograms (left) and bar graphs of quantification data showing the MFI of H2-Kb from two independent experiments (right) are shown. Data are means ± s.e.m. ($n = 3$ for each group). **$P < 0.01$, ****$P < 0.0001$; Student's $t$ test. **e** Expression of ligands for NK cell receptors (Qa-1b, Rae1, Mult1) on MLL-AF9 cells collected from WT or $Rag2^{-/-}$ mice.

advantage of $Rag2^{-/-}$ mice in this MLL-AF9 leukemia model was completely canceled by NK cell depletion (Fig. 4h). Taken together, we concluded that $Rag2^{-/-}$ mice have hyperactive NK cells, which play a major role in suppressing AML progression.

**Diverse and flexible immunogenicity of mouse AML cells.**
Finally, we assessed the relative importance of the innate and adaptive immune systems in the development of myeloid tumors

driven by other oncogenes. Similar to MLL-AF9-induced AML, the onset of RUNX1-ETO9a-induced AML was delayed in $Rag2^{-/-}$ mice compared to WT mice, and it became more evident with the treatment with DS-5272 (Fig. 5a). We then performed similar experiments using two MDS/AML cells that we established previously: cRAM (combined expression of RUNX1 and ASXL1 Mutations)[21] and cSAM (combined expression of SETBP1 and ASXL1 Mutations)[22,23] cells. In contrast to the results for MLL-AF9 and RUNX1-ETO9a-driven AML models, we observed earlier

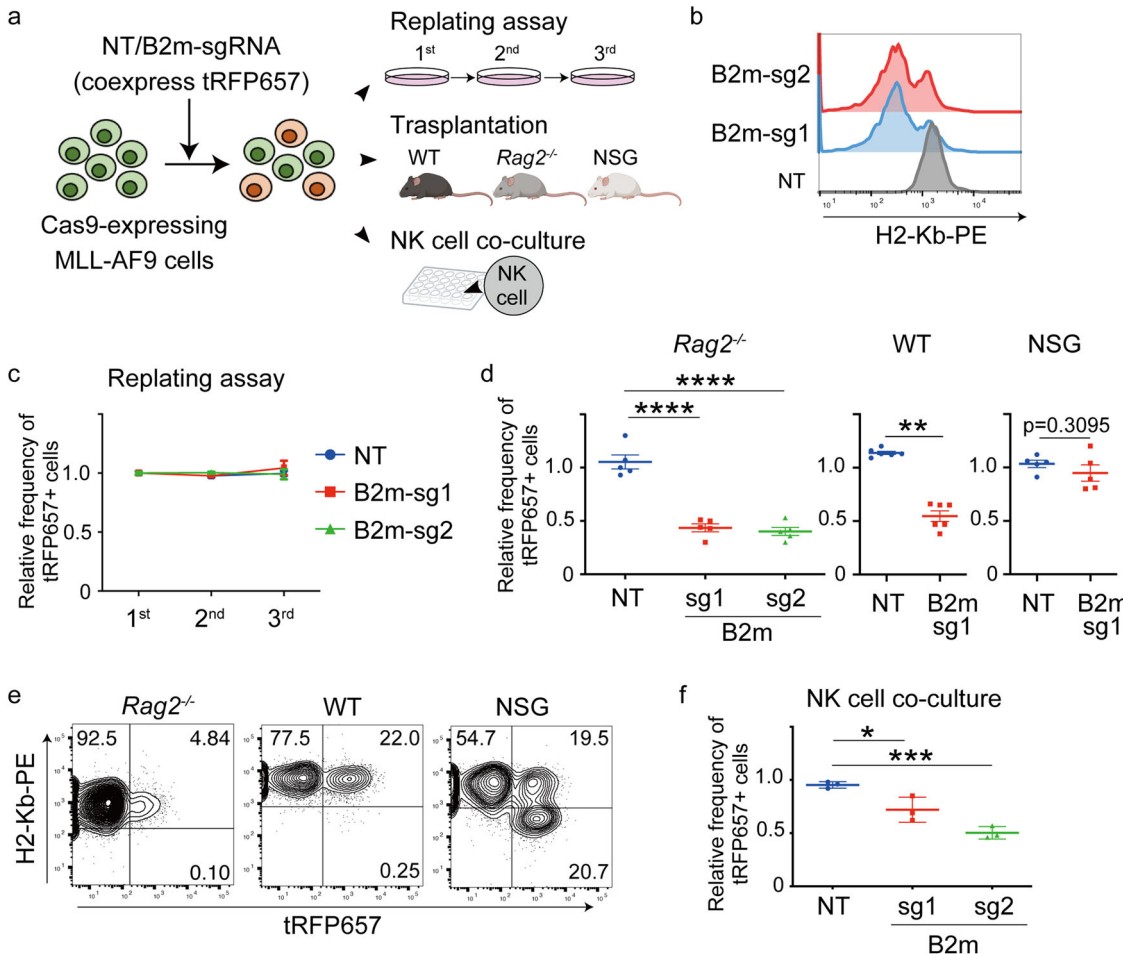

**Fig. 3 B2m depletion inhibits MLL-AF9-induced leukemogenesis. a** Experimental scheme used in **b**–**f** MLL-AF9 cells used in Fig. 1 were transduced with non-targeting (NT) sgRNA or sgRNAs targeting B2m coexpressing tRFP657. After transduction, the cells were cultured in methylcellulose (replating assay), transplanted into mice, or co-cultured with NK cells derived from spleens of WT mice. Figures were created partly with BioRender (https://app.biorender. com/). **b** Flow cytometry analysis of MHC Class I (H2-Kb) expression in MLL-AF9 cells after sgRNA transduction. **c**, **d** Changes of the ratio of tRFP657$^+$ (sgRNA-transduced) cells in MLL-AF9 cells in replating assay (**c**, $n = 3$) and transplantation assay (**d**, $Rag2^{-/-}$: $n = 5$, WT: $n = 6$, NSG: $n = 5$, male mice). Data are shown by the ratio to initial transduction and as mean ± s.e.m. **P < 0.01, ****P < 0.0001; one-way ANOVA with Tukey's multiple comparison test ($Rag2^{-/-}$ mice) or Mann–Whitney test (WT and NSG mice). **e** Representative FACS plots showing tRFP657 (x axis) and H2-Kb (y-axis) in B2m-sg1-transduced MLL-AF9 cells collected from WT, $Rag2^{-/-}$ and NSG mice. Note that the tRFP657$^+$ (B2m-sg1-transduced) cells from WT and $Rag2^{-/-}$ mice no longer lost H2-Kb expression. **f** Changes in the ratio of tRFP657$^+$ cells in MLL-AF9 cells co-cultured with NK cells derived from WT mice. Data are shown as mean ± s.e.m. ($n = 3$ per group). *P < 0.05, ***P < 0.001; one-way ANOVA with Dunnett's multiple comparisons test.

onset of cRAM cell-induced MDS/AML in $Rag2^{-/-}$ mice than WT mice (Fig. 5b), suggesting that cRAM cells are susceptible to adaptive immunity. Interestingly, two cSAM clones showed different leukemogenicity in WT and $Rag2^{-/-}$ mice. The original cSAM clone (cSAM-original), which was maintained through serial transplantations only in the "sublethally irradiated" mice, produced MDS/AML in $Rag2^{-/-}$ mice earlier than WT mice in both untreated and DS-5272-treated conditions (Fig. 5c). In contrast, another cSAM clone (cSAM-2020), which was maintained through serial transplantations in the "non-irradiated" mice, produced MDS/AML in WT mice earlier than $Rag2^{-/-}$ mice (Fig. 5d). These data suggest that each myeloid tumor has distinct immunogenicity, and it changes over time.

We then assessed expression of MHC-I, NK cell ligands and other immune checkpoint molecules in these myeloid tumors (Supplemental Fig. 3). However, none of them accounted for why some leukemia cells are resistant to innate or adaptive immune surveillance. Instead, we noticed that all leukemia cells that grow slowly in $Rag2^{-/-}$ mice (MLL-AF9, RUNX1-

ETO9a and cSAM-2020 cells) were maintained through serial transplantations in the "non-irradiated" mice. In contrast, other leukemias (cSAM-original and cRAM cells), which have relatively weak leukemogenicity, were maintained in the "sublethally irradiated" mice (Supplementary Table 1). Given the importance of immune systems in driving tumor evolution[24], the presence of normal immune system in the non-irradiated mice may promote the growth of AML cells that are resistant to adaptive immunity.

**Dynamic changes of immunogenicity of AML cells during tumor evolution.** To examine if AML cells change their immunogenicity during serial transplantations in immunologically normal mice, we generated new AML cells by transducing MLL-AF9 into mouse bone marrow progenitors and examined their immunogenic phenotypes over time (Fig. 6a). The newly generated MLL-AF9 cells (MLL-AF9-2021) produced AML more quickly in $Rag2^{-/-}$ mice than WT mice (Fig. 6b), indicating that

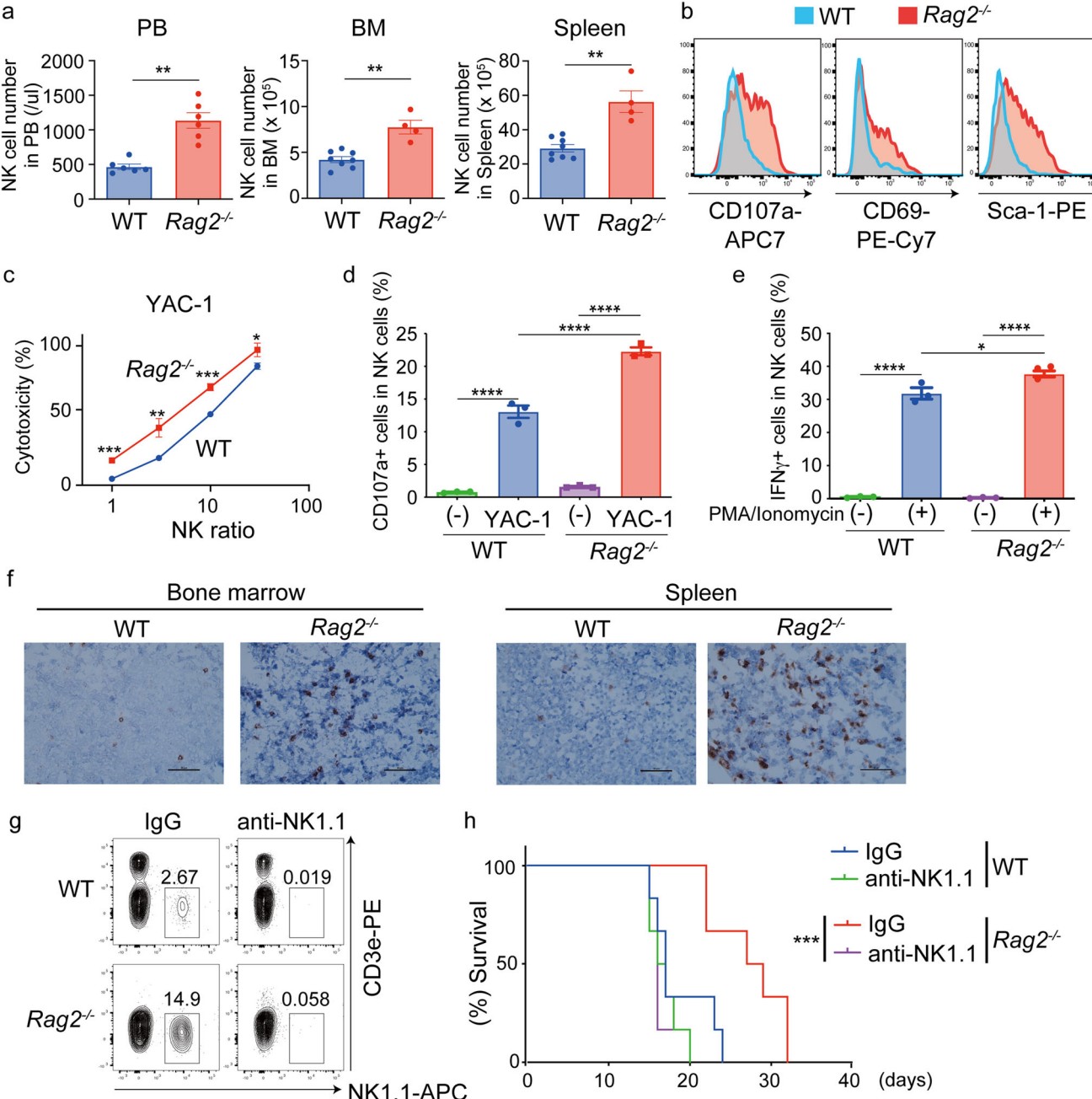

**Fig. 4 Rag2$^{-/-}$ mice have hyperactive NK cells. a** Number of NK cells in peripheral blood (PB), bone marrow (BM, both femurs and tibias) and spleen. Data are means ± s.e.m. (PB $n = 6$ per group, BM and spleen WT: $n = 8$, Rag2$^{-/-}$: $n = 4$, male mice). **$P < 0.01$; two-tailed Mann–Whitney test. **b** Expression of NK cell activation markers (CD107a, CD69, Sca-1) in BM-derived NK cells of WT and Rag2$^{-/-}$ mice. Representative FACS plots are shown. **c, d** NK cells were isolated from spleens of male WT and Rag2$^{-/-}$ mice and co-cultured with YAC-1 cells. Cytotoxicity at different leukemia cell-NK cell ratios (**c**) and expression of the degranulation marker CD107a in NK cells mixed with YAC-1 cells at a 1:1 ratio (**d**) are shown. Data are shown as means ± s.d (**c**) and means ± s.e.m (**d**) of triplicate wells. *$P < 0.05$, **$P < 0.01$, ***$P < 0.001$, ****$P < 0.0001$ two-tailed Student's t test (**c**) and Tukey's multiple comparisons test (**d**). Representative data of two independent experiments are shown. **e** Isolated NK cells were stimulated with PMA/ionomycin for 4 h, and IFNγ was measured by flow cytometry. Data are shown as means ± s.e.m. of triplicate wells. *$P < 0.05$, ****$P < 0.0001$; Tukey's multiple comparisons test. Representative data of two independent experiments are shown. **f** WT and Rag2$^{-/-}$ mice (male) were transplanted with MLL-AF9 cells and were sacrificed 11 days after transplantation to resect tibiae and spleens. Fixed bone marrow and spleen sections from WT and Rag2$^{-/-}$ mice were immunostained with anti-NK1.1 antibody to identify NK cells. Bars, 50 μm. **g, h** WT and Rag2$^{-/-}$ mice were treated with an anti-NK1.1 antibody or IgG control. The mice were then transplanted with MLL-AF9 cells. Efficient depletion of NK cells in mice by anti-NK1.1 antibody was confirmed by FACS (**g**). Kaplan–Meier survival curves of MLL-AF9-bearing mice in the presence or absence of NK cells are shown (**h**). ($n = 6$ for each group, female mice). ***$P < 0.001$; log-rank test.

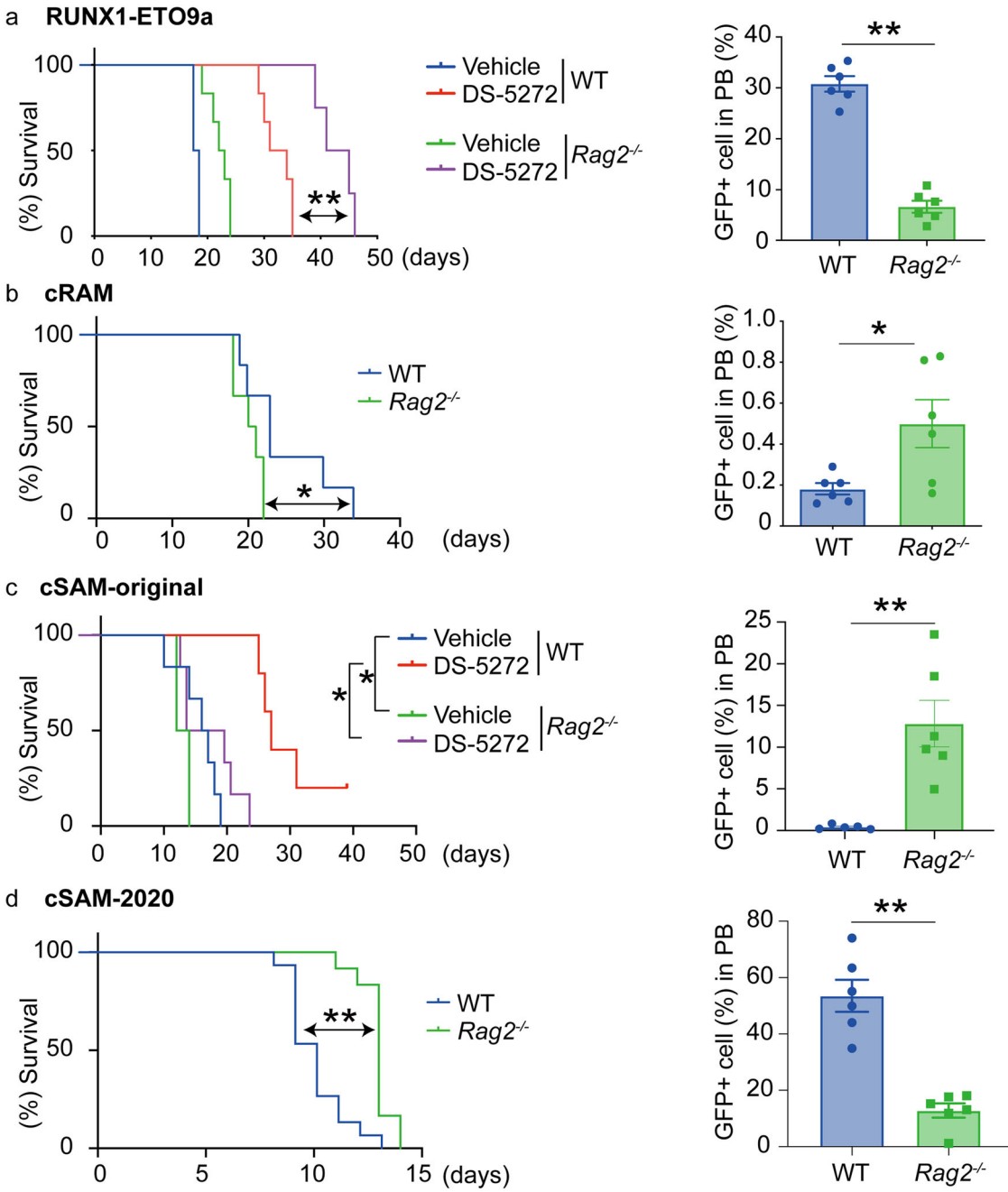

**Fig. 5 Diverse immunogenicity of myeloid tumors. a** RUNX1-ETO9a-expressing AML cells were transplanted into WT or $Rag2^{-/-}$ mice. The mice were then treated with vehicle control or DS-5272 (100 mg/kg, three times weekly for two weeks, starting three days after transplantation). (Left) Kaplan–Meier survival curves are shown. (WT/Vehicle: $n = 6$, WT/DS-5272: $n = 6$, $Rag2^{-/-}$/Vehicle: $n = 6$, $Rag2^{-/-}$/DS-5272: $n = 4$, male mice). **$P < 0.01$; log-rank test. (Right) Number of GFP$^+$ RUNX1-ETO9a cells in peripheral blood of WT and $Rag2^{-/-}$ mice 14 days after transplantation. Data are shown as means ± s.e.m. ($n = 6$ for each group, male mice). **$P < 0.01$; two-tailed Mann–Whitney test. **b** cRAM cells were transplanted into WT or $Rag2^{-/-}$ mice. (Left) Kaplan–Meier survival curves are shown. ($n = 6$ for each group, female mice). *$P < 0.05$; log-rank test. (Right) Number of GFP$^+$ cRAM cells in peripheral blood of WT and $Rag2^{-/-}$ mice 11 days after transplantation. Data are shown as means ± s.e.m. ($n = 6$ for each group, female mice). *$P < 0.05$; two-tailed Mann–Whitney test. **c** cSAM-original cells were transplanted into WT or $Rag2^{-/-}$ mice. The mice were then treated with vehicle control or DS-5272 (100 mg/kg, three times weekly for 3 weeks, starting three days after transplantation). (Left) Kaplan–Meier survival curves are shown. (WT/Vehicle: $n = 6$, WT/DS-5272: $n = 5$, $Rag2^{-/-}$/Vehicle: $n = 6$, $Rag2^{-/-}$/DS-5272: $n = 6$, female mice) *$P < 0.05$; log-rank test. (Right) Number of GFP$^+$ cSAM-original cells in peripheral blood of WT and $Rag2^{-/-}$ mice 10 days after transplantation. Data are shown as means ± s.e.m. (WT: $n = 5$, $Rag2^{-/-}$: $n = 6$, female mice). **$P < 0.01$; two-tailed Mann–Whitney test. **d** cSAM-2020 cells were transplanted into WT or $Rag2^{-/-}$ mice. (Left) Kaplan–Meier survival curves are shown. [WT: $n = 15$ (male/female=8/7), $Rag2^{-/-}$: $n = 12$ (male/female = 6/6)]. **$P < 0.01$; log-rank test. (Right) Number of GFP$^+$ cSAM-2020 cells in peripheral blood of WT and $Rag2^{-/-}$ mice 10 days after transplantation. Data are shown as means ± s.e.m. ($n = 6$ for each group, male mice). **$P < 0.01$; two-tailed Mann–Whitney test.

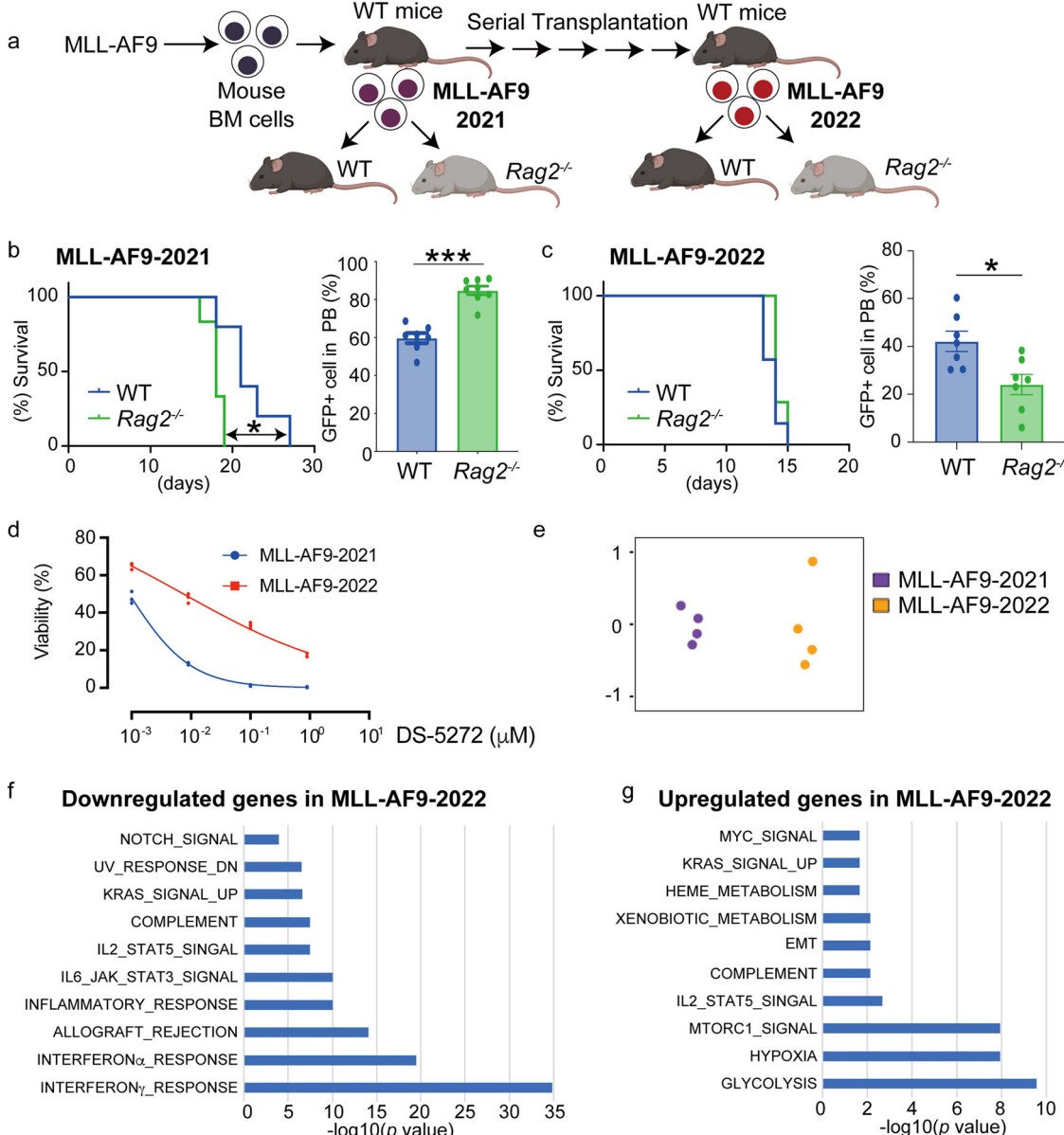

**Fig. 6 Flexible immunogenicity of myeloid tumors. a** Experimental scheme used in **b–g**. Mouse bone marrow progenitor cells were transduced with MLL-AF9 and were transplanted into sublethally irradiated WT mice. GFP+ MLL-AF9 cells were collected from moribund mice (MLL-AF9-2021) and were then transplanted into WT and $Rag2^{-/-}$ mice (**b**). Some MLL-AF9-2021 cells were subjected to serial transplantation with non-irradiated recipient mice to establish aggressive AML cells with strong repopulating ability (MLL-AF9-2022). The MLL-AF9-2022 cells were then transplanted into WT and $Rag2^{-/-}$ mice (**c**). Figures were created with BioRender (https://app.biorender.com/). **b** (Left) Kaplan–Meier survival curves are shown. ($n = 6$ for each group, male mice). **P < 0.01; log-rank test. (Right) Number of GFP+ MLL-AF9-2021 cells in peripheral blood of WT and $Rag2^{-/-}$ mice 14 days after transplantation. Data are shown as means ± s.e.m. (WT: $n = 7$, $Rag2^{-/-}$: $n = 8$, male mice). ***P < 0.001; two-tailed Mann–Whitney test. **c** (Left) Kaplan–Meier survival curves are shown. ($n = 7$ for each group, male mice). (Right) Number of GFP+ MLL-AF9-2022 cells in peripheral blood of WT and $Rag2^{-/-}$ mice 12 days after transplantation. Data are shown as means ± s.e.m. ($n = 7$ for each group, male mice). *P < 0.05; two-tailed Mann–Whitney test. **d** MLL-AF9-2021 and MLL-AF9-2022 cells were incubated with DS-5272 at the indicated concentration for 48 h. Cell viability assays were performed using the Cell Counting Kit-8. Data are shown as means ± standard deviation (SD) from three technical replicates. **e** Multi-Dimensional Scaling of RNA-Seq data. $n = 4$ per group, male mice. **f, g** GSEA for down- (**f**) and up- (**g**) regulated genes in MLL-AF9-2022 cells compared to MLL-AF9-2021 cells using the Hallmark collections of the GSEA MSigDB (http://software.broadinstitute.org/gsea/msigdb). The x-axis shows the P value (−log10).

they are susceptible to adaptive immunity. We then performed serial transplantations of MLL-AF9 cells into the non-irradiated recipient mice 5 times to generate aggressive AML cells with strong repopulating ability (MLL-AF9-2022). In contrast to the result of MLL-AF9-2021, MLL-AF9-2022 cells showed a tendency to develop leukemia more quickly in WT mice than $Rag2^{-/-}$ mice (Fig. 6c). Thus, it appears that MLL-AF9-2022 cells acquired

resistance to adaptive immunity during serial transplantations in non-irradiated recipient mice.

We then accessed molecular and phenotypic differences between the newly generated AML cells (MLL-AF9-2021) and aggressive AML cells generated through serial transplantations (MLL-AF9-2022). MLL-AF9-2022 cells became less sensitive to DS-5272, indicating that they acquire resistance to the cytotoxic

drugs (Fig. 6d). More interestingly, they showed distinct gene expression profiles (Fig. 6e, Supplementary Data 2). Genes related to interferon and inflammatory responses were significantly downregulated in MLL-AF9-2022 cells, which could account for their reduced immunogenicity to adaptive immunity (Fig. 6f). We also found that genes related to mTORC1, Hypoxia and Glycolysis were upregulated in MLL-AF9-2022 (Fig. 6g), indicating that the aggressive AML cells depend more on the mTORC1/HIF1 pathway and glycolysis. Thus, the immunogenicity as well as gene expression of myeloid tumors change dynamically during tumor evolution.

## Discussion

Rag2 is an essential component of the enzyme complex that initiates V(D)J gene rearrangement at the antigen receptor loci in lymphocyte development[25]. Consequently, $Rag2^{-/-}$ mice fail to produce mature B or T lymphocytes[26]. In contrast to T/B lymphocytes, NK cells do not require the Rag-mediated V(D)J recombination for their development. Nevertheless, about half of NK cells are derived from Rag-expressing lymphoid progenitors, giving rise to functionally heterogeneous NK cell populations[27]. Intriguingly, consistent with our data, a previous study showed that $Rag2^{-/-}$ NK cells exhibited an activated phenotype in vitro[27]. Although the $Rag2^{-/-}$ NK cells showed a diminished capacity to survive following transplantation and virus infection in the previous report[27], we did not observe the survival defect of $Rag2^{-/-}$ NK cells in our AML models. Consequently, the hyperactive NK cells in $Rag2^{-/-}$ mice showed enhanced antileukemia effect in several AML models. In addition, the absence of Treg cells in $Rag2^{-/-}$ mice could also result in the activation of NK cells to inhibit leukemogenesis[28]. Of note, a clinical study showed that SCID patients with RAG mutations also have activated NK cells, which is likely associated with the high rates of graft rejection after hematopoietic stem cell transplantation[29]. Collectively, these findings suggest that the hyperactive NK cells inhibit the engraftment and expansion of normal and malignant hematopoietic cells in Rag-deficient mice and humans.

Our study also revealed the flexible immunogenicity of myeloid tumors that could change with time and surrounding environments. The dynamic change of the immunogenicity of MLL-AF9 cells explains why previous studies using similar MLL-fusion leukemia models did not observe the delayed leukemia development in $Rag2^{-/-}$ mice compared to WT mice[18,30]. Importantly, expression of MHC-I and other immune checkpoint molecules did not predict the sensitivity of myeloid tumors to the specific immune system. For example, RUNX1-ETO9a cells express high level of MHC-I, which is supposed to promote T-cell activation, in fact produced AML more quickly in WT mice with functional T cells. In addition, cSAM-original and cSAM-2020 cells showed similar expression pattern of the ligands despite their different immunogenicity. Identification of biomarkers that predict responses of myeloid tumors to innate and adaptive immunity is an important future challenge. Notably, the aggressive AML cells established through multiple transplantations were still sensitive to NK cell-mediated tumor suppression. It is therefore tempting to speculate that NK cell-based immunotherapies could be effective to the refractory/relapsed human AMLs that survive chemotherapy in the presence of functional immune system.

In summary, we showed that certain subtypes of AML are particularly susceptible to NK cell-mediated antitumor immunity. The excellent therapeutic effect of the p53-activating drug in $Rag2^{-/-}$ mice suggest that combination of NK cell-based immunotherapy with p53-activating drug would be a promising treatment strategy for AML. Furthermore, our study highlighted the peculiar immune

phenotypes of $Rag2^{-/-}$ mice that lack functional lymphocytes but have hyperactive NK cells.

## Methods

**Mice and drug/antibody treatments.** Wild-type (WT) C57BL/6 J mice and C57BL/6J-Rag2em3Lutzy/J ($Rag2^{-/-}$) mice were purchased from Sankyo Labo Service corporation (Japan). NSG mice were purchased from Charles River Laboratories Japan. All experiments were performed with 8–12-week-old mice. All animal studies were approved by the Animal Care Committee of the Institute of Medical Science at the University of Tokyo (approval number: PA15-100, PA18-46), and were conducted in accordance with the Regulation on Animal Experimentation at the University of Tokyo based on International Guiding Principles for Biomedical Research Involving Animals. We have complied with all relevant ethical regulations for animal testing.

For drug studies, DS-5272 (Daiichi Sankyo) was dissolved in 0.5 w/v% Methyl Cellulose 400 Solution (Wako, Japan), and 100 mg/kg/day of DS-5272 was orally administered to mice three times a week for 2 or 3 weeks. For NK cell depletion, 200 μg of anti-NK1.1 antibody (BioXCell, clone PK136, #BE0036) was dissolved in *InVivo*Pure pH7.0 Dilution Buffer (BioXCell, #IP0070) and injected intraperitoneally once a week from one day before transplantation.

**Plasmid and viral transduction.** We used pMSCV-MLL-AF9-pgk-EGFP for MLL-AF9 expression[31]. For RUNX1-ETO expression, we cloned RUNX1-ETO9a into pMYs-IRES-EGFP vector. Retroviruses were produced by transient transfection of Plat-E packaging cells with retroviral constructs using the calcium-phosphate method[32]. Lentiviruses were produced by transient transfection of 293 T cells with viral plasmids along with gag-, pol-, and env-expressing plasmids [(pMD2.G #12259) and (psPAX #12260)] using the calcium-phosphate method[33]. Retrovirus transduction to the cells was performed using Retronectin (Takara Bio Inc, Otsu, Shiga, Japan).

**Mouse models for AML and MDS/AML.** MLL-AF9 cells were generated by transducing MLL-AF9 into murine bone marrow progenitor cells[31]. cSAM cells were generated by transducing SETBP1-D868N and ASXL1-E635RfsX15 into murine bone marrow progenitor cells[22]. RUNX1-ETO9a cells were generated by transducing RUNX1-ETO9a into murine fetal liver cells[33]. cRAM cells were generated by transducing RUNX1-S291fsX300 into mouse bone marrow progenitor cells derived from conditional knock-in mice expressing ASXL1-MT[21,34]. The oncogene-transduced cells were first transplanted intravenously into sublethally irradiated (525 cGy) recipient mice. The serial transplantation into sublethally irradiated recipient mice was repeated several times for cRAM cells and cSAM-original cells. MLL-AF9 cells, RUNX1-ETO9a cells, cSAM-2020 cells, and MLL-AF9-2022 cells were injected intravenously into non-irradiated recipient mice to generate AML and MDS/AML cells with strong repopulating ability. All these leukemia cells were transplanted intravenously ($1.0 \times 10^6$ cells/mouse) into non-irradiated recipient mice in the experiments to compare the survival of WT, $Rag2^{-/-}$ and NSG mice.

**Cell culture.** MLL-AF9 cells were cultured in MethoCult™ M3234 (STEMCELL Technologies, Bancouber, BC, Canada) or IMDM medium supplemented with 20% fetal bovine serum (FBS), with 10 ng/ml mouse stem cell factor (SCF), 10 ng/ml mouse granulocyte-macrophage colony-stimulating factor (GM-CSF), 10 ng/ml mouse interleukin-3 (IL-3) and 10 ng/ml mouse

interleukin-6 (IL-6) (R&D Systems, Minneapolis, MN). YAC-1 cells (TIB-160, ATCC, Manassas, VA, USA) were cultured in RPMI-1640 medium supplemented with 10% FBS. Murine NK cells were cultured in RPMI-1640 containing 10% FBS, 2-ME (50 μM), HEPES (10 mM), nonessential AA, sodium pyruvate, 10 ng/ml recombinant mouse IL-15 (Peprotech) and 100 U/ml mouse IL-2 (R&D Systems). 293 T cells (CRL-11268, ATCC) were cultured in DMEM media containing 10% FBS.

**B2m depletion using CRISPR/Cas9 system**. To generate sgRNA expression vectors for targeting mouse B2m, annealed oligos were inserted into pLKO5.sgRNA.EFS.tRFP657 vector (Addgene #57824) or lentiGuide-Puro vector (Addgene #52963)[35,36]. The expression vector for Cas9 (lentiCas9-Blast #52962) was also obtained from Addgene[35]. MLL-AF9 cells were infected with the virus for 24 h and were selected for stable expression of Cas9 using blasticidin (10 μg/ml). sgRNA-transduced MLL-AF9 cells were transplanted into mice or cultured in vitro. Initial transduction efficiency was analyzed at day 4 in culture, when tRFP657 expression was detected. Sequences for the-targeting (NT) control and sgRNAs targeting B2m are provided as follows: NT: 5' CGCTTCCGCGGCCCGTTCAA 3',
B2m-sg1: 5'-ACTCACTCTGGATAGCATAC-3',
B2m-sg2: 5'-ATTTGGATTTCAATGTGAGG-3'.

**Flow cytometry**. Bone marrow cells were obtained by crushing femurs and tibias in PBS containing 2% FBS. Red blood cells were removed using RBC lysis buffer. Cells were then stained by fluoro-conjugated antibodies for 30 min at 4 °C. After staining, cells were washed with cold PBS several times, and were resuspended with PBS containing 2% FBS. Cells were analyzed on a FACS Verse or FACS CantoII, and were sorted with a FACSAria (BD Biosciences, San Jose, CA, USA). Apoptosis analysis (AnnexinV-PE, BD Pharmingen™, Cat#556421, 1:100) were performed according to the manufacturer's protocol. The data were analyzed using FlowJo software (Treestar, Inc., San Carlos, CA). 4′,6-diamidino-2-phenylindole (DAPI) (BioLegend, Cat#422801, 1:3000) was used to exclude dead cells. Antibodies used in this study are provided in Supplementary Table 2. Gating strategies are provided in Supplemental Fig. 4.

**NK cell co-culture assay**. NK cells (CD49b$^+$CD3e$^-$) were isolated from spleens of WT or Rag2$^{-/-}$ mice using FACS Aria. For cytotoxicity assay, YAC-1 cells are labeled with CellTrace Far Red (Thermo Fisher Scientific, C34564) according to the manufacturer's protocol. $5 \times 10^3$ labeled YAC-1 cells were seeded in 96-well plate with freshly sorted NK cells at different effector: target ratio (0:1, 1:1, 3:1, 10:1, 30:1). After 4 h of culture in 37 °C, cells were washed and stained with AnnexinV and DAPI. The percentage of AnnexinV positive cells in labeled YAC-1 cells (CellTrace Far Red$^+$) was defined by flow cytometry. Cytotoxicity (%) was calculated as follows: Cytotoxicity (%) = (AnnexinV$^-$DAPI$^-$ spontaneous (%) - AnnexinV$^-$DAPI$^-$ experimental (%))/AnnexinV$^-$DAPI$^-$ spontaneous (%) × 100 (%).

For cytotoxicity assay against sgRNA-transduced MLL-AF9 cells, NK cells were pre-activated by 5 days culture in the presence of IL-2 and IL-15. The activated NK cells were co-cultured with sgRNA (coexpress tRFP657)-transduced Cas9-expressing MLL-AF9 cells. After 24 h, percentage of tRFP657$^+$ in MLL-AF9 cells were analyzed by flow cytometry.

For degranulation assay, $5 \times 10^4$ YAC-1 cells were co-cultured with same numbers of freshly sorted NK cells (E:T ratio = 1:1) in the presence of anti-CD107a-APC antibody at 37 °C. After 2 h, GolgiStop (BD) was added, and the cells were incubated another 4 h at 37 °C. Following total 6-hour incubation, cells were stained

with anti-NK1.1-PE and anti-CD3e-PE-Cy7 antibodies and were analyzed by flow cytometry.

For intracellular IFNγ staining, $5 \times 10^4$ freshly sorted NK cells were incubated with PMA (50 ng/ml), ionomycin (10 μg/ml) and GolgiStop for 4 h at 37 °C. After washing, cells were stained with anti-NK1.1-PE and anti-CD3e-PE-Cy7. After then, cells were fixed and permeabilized using Cytofix/Cytoperm™ Kit (BD) following the manufacturer's protocol followed by staining with anti-IFNγ-APC. IFNγ expression in NK cells was analyzed by flow cytometry.

**RNA-seq analysis**. Total RNA was extracted from $3.0 \times 10^6$ GFP$^+$ leukemia cells using RNeasy Mini Kit (Qiagen). mRNA was purified from total RNA using poly-T oligo-attached magnetic beads. Pair-end sequencing FASTQ files were aligned to the mouse reference genome (mm10) using HISAT2[37] on Galaxy platform (https://usegalaxy.org). For the RNA-seq shown in Fig. 2, Raw gene counts were obtained from read alignments on Galaxy history. TPM were calculated from Raw gene counts. After filtering out low-expression genes with TPM lower than 1, all TPM values were Z score transformed using genefilter (v1.76.0) for generating hierarchical clustering and drawing heatmap using gplots (v3.1.1). Differential expression was analyzed with the linear model using limma (v3.46.0)[38]. Genes with false discovery rate (FDR) < 0.05 adjusted by the Benjamini–Hochberg method and logFC > 0.5 were considered significant differentially expressing genes (DEGs), which were further enrolled in gene ontology analysis using clusterProfiler (v3.18.1)[39]. For the RNA-seq shown in Fig. 6, raw gene counts were derived from the read alignments by Rsubread.1 (v2.12.3)[40] and further transferred into count per million (CPM) by edgeR2 (v3.40.2)[41]. After filtering out low-expression genes with CPM lower than 1, all CPM values were log2 transformed for generating unsupervised clustering dendrograms and heatmaps. Differential expression was analyzed with the linear model using limma3 (v3.54.2)[38]. Genes with false discovery rate (FDR) < 0.05 adjusted by the Benjamini-Hochberg method were considered DEGs. Gene set enrichment analysis (GSEA) was performed using the GSEA tool from the Broad Institute (http://software.broadinstitute.org/gsea/)[42].

**Histological analysis**. WT and Rag2$^{-/-}$ mice were transplanted with MLL-AF9 cells and were sacrificed 11 days after transplantation to resect tibiae and spleens. Bone marrows were flushed out using 21-gauge needle attach to a 1 ml syringe containing PBS. The bone marrows and spleens were fixed with 4% paraformaldehyde in PBS and were immunostained with anti-NK1.1 antibody [Klrb1c/CD161c (E6Y9G) Rabbit mAb #39197, Cell Signaling Technology]. Sections were analyzed using a microscope (Evos FL Auto 2, Thermo Fisher Scientific).

**Cell viability assay**. The cytotoxicity of DS-5272 was measured by the WST-8 assay. The MLL-AF9-2021 and MLL-AF9-2022 cells were seeded in 96-well plates at a density of $1 \times 10^4$ cells/well in 0.1 mL medium, followed by treatment with various concentrations of DS-5272 for 48 h. Then, 8 μL of the Cell Counting Kit-8 were added into each well. After incubation at 37 °C for 4 h, the plates were mixed and the absorbance at 450 nm was measured with a microplate reader (CLARIOstar Plus, BMG LABTECH, Ortenberg, GER). The relative cell viability was expressed as the ratio of the absorbance of each treatment group against those of the corresponding untreated control group.

**Statistics and reproducibility**. Statistical analyses were performed by the unpaired, two-tailed Student's *t* test, Mann–Whitney rank sum test, one-way ANOVA with Dunnett's or Tukey–Kramer's post hoc test as indicated in figure legends. The survival distributions were compared by the log-rank test. GraphPad Prism 9 was used for these statistical analyses, and the exact *p*-value for each comparison is provided as Supplementary Table 3. No specific statistical methods were used to predetermine the sample size.

## Data availability

RNA-seq data were deposited in the NCBI's Gene Expression Omnibus (GEO accession numbers GSE245358 and GSE245359). The source data for the graphs has been provided with the manuscript as Supplementary Data 3.

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

## Acknowledgements

We thank Akiho Tsuchiya for their expert technical assistance. We thank Hiroshi Kawamoto for the helpful discussions. We thank Daiichi Sankyo Corporation for providing DS-5272. We also thank the Flow Cytometry Core and the Mouse Core at The Institute of Medical Science, The University of Tokyo. This work was supported by Grant-in-Aid for Scientific Research (A) (20H00537, T.K.), Grant-in-Aid for Scientific Research (B) (19H03685 and 22H03100, S.G.), Grant-in-Aid for Challenging Research (Exploratory) (22K19540, S.G.), AMED under Grant Number (21ck0106644h0001 and 22ama221514s0201, S.G.), a research grant from The Japanese Society of Hematology (T.K., S.G.) and a research grant from Astellas Pharma (S.G.).

## Author contributions

E.S. designed and performed experiments, analyzed the data and wrote the paper. J.L. designed and performed experiments and analyzed the data. Y.H. performed experiments and analyzed the data. K.I., S.A., T. Fukushima., M.T., S.S., W.Z., K.Y., K.C.K., T. Kawase, T.S., T.Y., Y.K., Y.I., T. Fukuyama, Y.T., Y.E. analyzed the data, S.Y. provided the mice, T.D. and Y.O. performed histological analysis, T.Kitamura and S.G. conceived the project, designed experiments, analyzed the data and wrote the paper.

## Competing interests

The authors declare no competing interests.

**Additional information**

