## [Peer Review File · Communications Biology]

Reviewers' comments:

Reviewer #1 (Remarks to the Author):

The manuscript by Sugimoto et al is interesting and attempts to understand the role of NK cells in acute myeloid leukemia. They utilize mouse transplantation models of acute myeloid leukemia to compare leukemogenesis in WT and Rag2^{-/-} mice (and in one experiment also NSG mice). They found that following serial transplantation of transfected AML cells to non-irradiated recipient mice, the AML cells grew less aggressively in Rag2^{-/-} mice compared with WT mice, and responded better to p53-MDM2 interaction inhibition in Rag2^{-/-} mice.

Comments and concerns:

1. I suggest to already in the abstract explain that the reduced growth of AML cells in Rag2^{-/-} was only seen after serial passage of leukemic cells through non-irradiated WT mice.
2. A key question that has not been properly addressed is which features that are altered in leukemic cells following passage through non-irradiated WT mice, that alters their ability to grow in WT vs Rag2^{-/-} mice. To address this I suggest:
 - A) Gene expression analysis of the leukemic cells that have been passaged or not through non-irradiated mice
 - B) In vitro assays using these leukemic cells where their sensitivity towards NK cell killing and T cell killing is compared
3. The study also implies that leukemic cells that have been passaged or not through non-irradiated mice differ in sensitivity towards DS-5272.
 - A) Is it correct for all cell lines that they become more sensitive to DS-5272 in Rag2^{-/-} mice compared with WT mice after passage through non-irradiated mice?
 - B) Do cell lines that have been passaged or not through non-irradiated mice differ in in vitro sensitivity to DS-5272? Do you have any idea of an explanation to this feature in that case?
4. I would say that it is well accepted already in the scientific community that Rag2^{-/-} mice have more active NK cells. Hence, I would modify the final statement of the discussion.

Reviewer #2 (Remarks to the Author):

In the manuscript "Hyperactive NK cells in Rag2 knockout mice inhibit the development of acute myeloid leukemia", Sugimoto et al. demonstrated that retarded outgrowth of MLL-AF9 AML cells in Rag2 knockout mice (no B/T cells) as compared to WT mice is mediated by hyperactive NK cells. The authors also showed that MLL-AF9 AML cells are resistant to adaptive immunity (but still sensitive to innate or NK cell immunity) due to serial transplantation in non-irradiated mice, while MLL-AF9 AML cells remain sensitive to adaptive immunity when serially transplanted in irradiated mice. The manuscript is well written and the conclusions are supported by data and clear Figures.

Comments

[1] It is unclear how the MLL-AF9 AML cells used in Figure 1 have been generated. How often have the AML cells been serially transplanted in (non-)irradiated mice? This is important, since Figure 5 shows that MLL-AF9 that are serially transplanted in non-irradiated mice are resistant to adaptive immunity but still sensitive to innate or NK cell immunity (slower outgrowth in Rag2 knockout mice than WT mice), while MLL-AF9 AML cells remain sensitive to adaptive immunity when serially transplanted in irradiated mice (slower outgrowth in WT mice than Rag2 knockout mice).

[2] The authors should use MLL-AF9 AML cells to compare the functionality of NK cells between WT and Rag2 knockout mice or explain why YAC-1 cells are used in Figure 4c & 4d.

[3] Figures 1b, 3d & 3e show that retarded outgrowth of MLL-AF9 AML cells in WT mice as compared to NSG mice is mediated by NK cells. Why does NK cell depletion in WT mice (Figure 4g) not lead to accelerated growth of MLL-AF9 AML cells?

[4] It is unclear which NK cells have been used in the coculture assay in Figure 3f.

[5] The authors should describe how male and female mice have been distributed over the different groups in the various experiments, since tumor growth can be different in male and female mice.

Reviewer #3 (Remarks to the Author):

COMMSBIO-22-3184-T

The authors of the manuscript "Hyperactive NK cells in Rag2 knockout mice inhibit the development of acute myeloid leukemia" present data showing that Rag2^{-/-} mice show delay in development of MLL-AF9 induced AML, that p53-activating drug showed stronger antileukemia effect on AML in Rag2^{-/-} mice than C57BL/6 mice, NK cell depletion accelerated AML progression in Rag2^{-/-} mice and that Rag2^{-/-} mice have hyperactive NK cells with the enhanced antileukemia immunity. This is in general a well written manuscript. However clarification of major questions would improve the quality of this manuscript.

Figure 1. 1. The longest survival can be observed in Rag2^{-/-} mice, however the survival differences are small. The experiment with the p53-MDM2 interaction inhibitor, DS-5272, show an anti-leukemic effect in Rag2^{-/-} mice. Please clarify the connection to figure one. What matches better here is what is shown in Figure 4f,g, that upon depletion of NK cells in the Rag2^{-/-} mice the delay in leukemic mortality is abrogated. However, here in Figure 4g again not a single mouse is cured in the Rag2^{-/-} mice. Does reduced numbers of MLL-Af9 leukemia allow survival of leukemic Rag2^{-/-} mice?

2. Secondly, how do the authors explain a survival benefit of just a few days (5-6?) in the RAG2^{-/-} mice compared to the NSG mice, with not a single mouse cured?

3. Do NK cells not reach all AML cells e.g. in the BM? Figure 1c and d only show spleen analysis. This can be addressed experimentally by performing e.g. histological analysis.

4. The authors conclude that "Rag2^{-/-} mice have the unexpected tumor suppressive environment for MLL-AF9-induced AML". The authors can consider changing this statement, and rephrase it, since there is evidence in the literature, not referenced here, that NK cells are anti-leukemic e.g. PMID: 29154208

5. Figure 2- the authors describe that "AML cells in Rag2^{-/-} mice show inflammatory phenotype". However the data in Figure 2a-c is only descriptive, a verification of expression of key genes is missing. The connection of Fig. 2a-c and the following experiments looking at expression of MHC-I and other NK cell receptors is not immediately clear.

6. Figure 4 the information in Figure 4a is not surprising and may be moved to the supplementary data.

Reviewer #1

We truly appreciate the insightful and constructive comments from the reviewer 1. The followings are point-by-point responses to the reviewer's comments and concerns.

Comment #1

I suggest to already in the abstract explain that the reduced growth of AML cells in Rag2^{-/-} was only seen after serial passage of leukemic cells though non-irradiated WT mice.

Response to Comment #1

Thank you for the suggestion. We changed the abstract accordingly.

Comment #2

A key question that has not been properly addressed is which features that are altered in leukemic cells following passage through non-irradiated WT mice, that alters their ability to grow in WT vs Rag2^{-/-} mice. To address this I suggest: A) Gene expression analysis of the leukemic cells that have been passaged or not through non-irradiated mice B) In vitro assays using these leukemic cells where their sensitivity towards NK cell killing and T cell killing is compared

Response to Comment #2

We appreciate this reviewer's comment. According to this reviewer's suggestion, we performed RNA-seq analysis using newly generated MLL-AF9 cells (MLL-AF9-2021) and aggressive MLL-AF9 cells generated through 5 times serial transplantations in non-irradiated mice (MLL-AF9-2022). Interestingly, they showed distinct gene expression profiles (Figure 6e). Genes related to interferon and inflammatory responses were significantly downregulated in MLL-AF9-2022 cells (Figure 6f), which could account for their reduced immunogenicity to adaptive immunity. We also found that genes related to mTORC1, Hypoxia and Glycolysis were upregulated in MLL-AF9-2022 cells, indicating the important role of the mTORC1/HIF1-mediated glycolysis in the aggressive AML (Figure 6g). Thus, the RNA-seq analysis clearly revealed several features of MLL-AF9-2022 cells with strong leukemogenicity, including reduced interferon/inflammatory signals and activated mTORC1/HIF pathway. We added these results to the revised manuscript. Thank you again for the great suggestion!

Comment #3

The study also implies that leukemic cells that have been passaged or not through non-irradiated mice differ in sensitivity towards DS-5272. A) Is it correct for all cell lines that they become more sensitive to DS-5272 in Rag2^{-/-} mice compared with WT mice after passage though non-irradiated mice? B) Do cell lines that have been passaged or not though non-

irradiated mice differ in in vitro sensitivity to DS-5272? Do you have any idea of an explanation to this feature in that case?

Response to Comment #3

We assessed the sensitivity of newly established MLL-AF9 cells (MLL-AF9-2021) and those obtained after serial transplantations to DS-5272 *in vitro*. The MLL-AF9 cells obtained after serial transplantations were not as sensitive as MLL-AF9-2021 to DS-5272, indicating that they became more aggressive and more resistant to drug treatment. It is therefore likely that the excellent therapeutic effect observed in Fig.1 is not because the MLL-AF9 cells are sensitive to DS-5272, but because they are sensitive to NK cell-mediated cytotoxicity. We added the data to the revised manuscript (Figure 6d).

Comment #4

I would say that it is well accepted already in the scientific community that Rag2^{-/-} mice have more active NK cells. Hence, I would modify the final statement of the discussion.

Response to Comment #4

Thank you for the suggestion. We revised the final sentence accordingly.

Reviewer #2

We truly appreciate the insightful and constructive comments from the reviewer 2. The followings are point-by-point responses to the reviewer's comments and concerns.

Comment #1

It is unclear how the MLL-AF9 AML cells used in Figure 1 have been generated. How often have the AML cells been serially transplanted in (non-)irradiated mice? This is important, since Figure 5 shows that MLL-AF9 that are serially transplanted in non-irradiated mice are resistant to adaptive immunity but still sensitive to innate or NK cell immunity (slower outgrowth in Rag2 knockout mice than WT mice), while MLL-AF9 AML cells remain sensitive to adaptive immunity when serially transplanted in irradiated mice (slower outgrowth in WT mice than Rag2 knockout mice).

Response to Comment #1

The original MLL-AF9 cells were generated long time ago, and were maintained in mice through serial transplantations (more than 10 times). We added this explanation into the revised manuscript.

Comment #2

The authors should use MLL-AF9 AML cells to compare the functionality of NK cells between

WT and Rag2 knockout mice or explain why YAC-1 cells are used in Figure 4c & 4d.

Response to Comment #2

In the coculture assay in Figure 4c & 4d, we used **freshly isolated** NK cells from WT or Rag2 knockout mice. Because the freshly isolated NK cells were less active than the precultured NK cells, neither WT nor Rag2-deficient NK cells showed efficient cytotoxicity to MLL-AF9 cells. Therefore, we decided to use YAC-1 cells that are very sensitive to NK cells, even to the freshly isolated ones.

Comment #3

Figures 1b, 3d & 3e show that retarded outgrowth of MLL-AF9 AML cells in WT mice as compared to NSG mice is mediated by NK cells. Why does NK cell depletion in WT mice (Figure 4g) not lead to accelerated growth of MLL-AF9 AML cells?

Response to Comment #3

As the reviewer pointed out, NK cell depletion did not accelerate the development of AML in WT mice, probably because WT NK cells do not have sufficient activity to suppress MLL-AF9-driven leukemogenesis in this experimental setting. Please also note that NK cell depletion tended to promote AML development even in WT mice, although it did not reach the statistical significance.

Comment #4

It is unclear which NK cells have been used in the coculture assay in Figure 3f.

Response to Comment #4

We used WT NK cells in this experiment. We added the explanation to the revised manuscript. Thank you for the suggestion.

Comment #5

The authors should describe how male and female mice have been distributed over the different groups in the various experiments, since tumor growth can be different in male and female mice.

Response to Comment #5

We added the information in the Figure legends in the revised manuscript. We understand the possible influence of sex difference on tumor growth, so male/female distribution for recipient mice was same between the groups in all experiments. Thank you for the suggestion.

Reviewer #3

We truly appreciate the insightful and constructive comments from the reviewer 3. The followings are point-by-point responses to the reviewer's comments and concerns.

Comment #1

Figure 1. 1. The longest survival can be observed in Rag 2^{-/-} mice, however the survival differences are small. The experiment with the p53-MDM2 interaction inhibitor, DS-5272 , show an anti-leukemic effect in Rag 2^{-/-} mice. Please clarify the connection to sifure one. What matches better here is what is shown in Figure 4f,g, that upon depletion of NK cells in the Rag2^{-/-} mice the delay in leukemic mortality is abrogated. However, her in Figure 4 g again not a single mouse is cured in the Rag 2^{-/-} mice. Does reduced numbers of MLL-Af9 leukemia allow survival of leukemic Rag 2^{-/-} mice?

Response to Comment #1

Thank you for the insightful comment. Yes, it appears that NK cells can inhibit leukemogenesis efficiently only when mice have low numbers of leukemia cells. Therefore, it would be important to combine the NK cell therapy with molecular-targeted therapies (e.g., p53-activating drug) that can reduce the leukemia burden in future clinical trials.

Comment #2

Secondly, how do the authors explain a survival benefit of just a few days (5-6?) in the RAG2^{-/-} mice compared to the NSG mice, with not a single mouse cured ?

Response to Comment #2

NK cells showed only limited anti-leukemia effects in mice with abundant leukemia cells.

Comment #3

Do NK cells not reach all AML cells e.g. in the BM ? Figure 1c and d only show spleen analysis. This can be addressed experimentally by performing e.g. histological analysis.

Response to Comment #3

We appreciate this reviewer's suggestion. We performed histological analysis using bone marrow and spleen collected from WT and Rag2 knockout mice with MLL-AF9 leukemia cells. As expected, we observed the substantial increase of NK cells in both bone marrow and spleen in Rag2^{-/-} mice, which confirmed the enhanced NK cell-mediated tumor suppression in Rag2^{-/-} mice. We added the data to the revised manuscript (Figure 4f).

Comment #4

The authors conclude that "Rag2^{-/-} mice have the unexpected tumor suppressive environment for MLL-AF9-induced AML". The authors can consider changing this statement, and rephrase it, since there is evidence in the literature, not referenced here, that NK cells are anti-leukemic e.g. PMID: 29154208

Response to Comment #4

We believe the finding is unexpected for many, if not all, readers. Therefore, we would like to keep this sentence in the revised manuscript. Instead, we added the following sentence in the same paragraph and cited several references including the given one.

“Given that several studies have shown the antitumor effect of NK cells against MLL-fusion leukemia, these results also suggest the involvement of NK cells, which are retained in Rag2^{-/-} mice, in suppressing the development of AML.

Comment #5

Figure 2- the authors describe that “AML cells in Rag2^{-/-} mice show inflammatory phenotype”. However the data in Figure 2a-c is only descriptive, a verification of expression of key genes is missing. The connection of Fig. 2a-c and the following experiments looking at expression of MHC-I and other NK cell receptors is not immediately clear.

Response to Comment #5

As the reviewer pointed out, we did not confirm the inflammatory phenotype of AML cells derived from Rag2 knockout mice by other assays. Moreover, whether the inflammatory phenotype directly induced MHC-I upregulation is not clear. Therefore, we decided to tone down our description. Thank you for the suggestion.

Interestingly, our new RNA-seq analysis using newly generated (MLL-AF9-2021) and post-transplant (MLL-AF9-2022) AML cells revealed the reduced expression of inflammation/interferon-associated genes in the post-transplant aggressive AML cells (Figure 6f). These data indicated again the involvement of intracellular inflammation to regulate the sensitivity of AML cells to anti-tumor immunity.

Comment #6

Figure 4 the information in Figure 4a is not surprising and may be moved to the supplementary data.

Response to Comment #6

The data may not be surprising for the experts in the field, but we think they would be interesting to many (non-expert) readers. We therefore would like to keep them as part of main figures.

REVIEWERS' COMMENTS:

Reviewer #1 (Remarks to the Author):

The authors have addressed all comments satisfactorily.

Reviewer #2 (Remarks to the Author):

All my comments have been accurately addressed by the authors.

Reviewer #3 (Remarks to the Author):

Comments have been addressed.

No further comments.